# Influence of Heat Treatment on the Chemical, Physical, Microbiological and Sensorial Properties of Pork Liver Pâté as Affected by Fat Content

**DOI:** 10.3390/foods12122423

**Published:** 2023-06-20

**Authors:** Zuzana Lazárková, Alena Kratochvílová, Richardos Nikolaos Salek, Zdeněk Polášek, Ladislav Šiška, Markéta Pětová, František Buňka

**Affiliations:** 1Department of Food Technology, Faculty of Technology, Tomas Bata University in Zlin, 760 01 Zlin, Czech Republic; a_jedounkova@utb.cz (A.K.); rsalek@utb.cz (R.N.S.); zpolasek@utb.cz (Z.P.); siska@utb.cz (L.Š.); 2Laboratory of Food Quality and Safety Research, Department of Logistics, Faculty of Military Leadership, University of Defence, 662 10 Brno, Czech Republic; marketa.petova@unob.cz (M.P.); frantisek.bunka@gmail.com (F.B.)

**Keywords:** pork liver pâté, pasteurisation, sterilisation, fat level, lipid oxidation, texture, viscoelastic properties

## Abstract

The novelty of this study is the investigation of the effect of different heat treatments and, simultaneously, the effect of different fat levels on the quality of pork liver pâtés. Hence, this study aimed to evaluate the effect of heat treatment and fat content on selected properties of pork liver pâté. For this purpose, four batches of pâtés with two different fat contents (30 and 40% (*w*/*w*)) and two different heat treatments (pasteurisation: target temperature 70 °C, holding time of 10 min; sterilisation: target temperature 122 °C, holding time of 10 min) were manufactured. Chemical (pH, dry matter, crude protein, total lipid, ammonia, and thiobarbituric acid reactive substances (TBARS)), microbiological, colour, textural, rheological, and sensory analyses were performed. Both different heat treatment and fat content affected most of the parameters observed. Sterilisation ensured the commercial sterility of the manufactured pâtés, resulting in elevated TBARS values, hardness, cohesiveness, gumminess and springiness, and increased rheological parameters (G′, G″, G*, and η*), as well as colour changes (decrease in L* and increase in a*, b*, and C* values) and deterioration of appearance, consistency, and flavour also being detected (*p* < 0.05). Higher fat content caused similar variations in the textural and viscoelastic properties, i.e., the increase in hardness, cohesiveness, gumminess and springiness, and also in G′, G″, G*, and η* (*p* < 0.05). However, the colour and sensorial parameters changed in different ways compared to the changes induced by the sterilisation effect. Overall, the observed changes might not be desirable for some consumers and further research would be appropriate to improve especially the sensorial attributes of sterilised pork liver pâtés.

## 1. Introduction

Liver pâté is a product with important gastronomic traditions and highly appreciated sensorial properties that is consumed throughout the world (in Europe, mainly Denmark, France, Germany, and Spain). It can be manufactured from various types of meat (pork, beef, poultry, or fish) [1,2]. Pork liver pâté (PLP) can be defined as a paste produced using pork meat, liver, backfat, liquid (broth or water), seasoning, and auxiliary compounds (e.g., potassium phosphate, sodium chloride, sodium ascorbate, sodium caseinate, sodium nitrite, globin, etc.) that has been pasteurised or sterilised. For the packaging, sausage casings, glass, or metal containers are commonly used. Meat pâtés are spreadable emulsion-type meat products with a fat content of 25–50% [3,4,5]. The quality of these foods is related to proper mixing of fat, water, and soluble protein without fat separation after production and/or during storage [6,7].

Fat content plays an important role in the PLP quality. It has a considerable effect on the nutritional value (energy content), technological characteristics, as well as sensorial properties (tenderness, juiciness, and flavour). Furthermore, fat can interact with other ingredients contained in pâté and it improves consistency, overall taste, and consumers’ acceptance [1,3,8]. Due to the chemical composition (high amount of fat and iron, together with low content of natural antioxidants) and its manufacturing process (mincing and cooking), liver pâté is predisposed to oxidation damage [2,9]. 

The degradation of polyunsaturated fatty acids and the generation of free radicals are involved in lipid oxidation. The processes mentioned above lead to the deterioration of proteins, the oxidation of heme pigments, and the generation of rancid odours [10]. Oxidation affects both lipid and protein components, including meat pigments. These reactions can result in deterioration in quality and reduction in shelf life in terms of flavour, texture, colour, and nutritional value. Furthermore, some of the oxidation products could be toxic compounds that are harmful to human health and can implicate some human pathologies, including atherosclerosis, cancer, inflammation, and aging processes [2,11,12]. Hence, lipid oxidation is assumed as a major factor reducing the quality and acceptability of fat-containing muscle foods [10].

As mentioned above, PLP can be treated with either pasteurisation or sterilisation. Pasteurised products (usually treated by temperature 70–80 °C with holding time of several min) generally suffer from less severe quality changes; however, such foods must be stored in the refrigerator. Refrigeration storage has high requirements for transportation and storage conditions. Hence, the use of heat sterilisation can be used to ensure the safety of packaged low-acid food products (with a pH value greater than 4.6), such as PLP [13,14,15]. Sterilised products (treated by temperature 115–125 °C with a holding time of several min) are expected to have a shelf life of at least two years from the date of processing and can be stored at room temperature [16]. Consequently, these foodstuffs can be transferred overseas, are exploitable in areas with an imperfect or not available cold chain, and, last but not least, can be used in crisis situation feeding [15,16,17]. 

Retort processing is the technology most commonly used to sterilise food in different kinds of containers, usually cans. Commercially sterilised meat products are generally considered shelf-stable and are classified as “canned” or “commercially sterile” [18]. Retorting provides safety of meat products by eliminating virtually all microorganisms and their spores, capable of proliferating and/or producing toxins, as well as spoilage microorganisms [13,14].

However, due to extreme processing conditions (given by the combination of temperature and time), both the nutritional and sensory quality of the food deteriorate significantly. Quality changes occur in proteins, lipids, as well as vitamins and minerals [18]. Changes occurring in meat proteins affect texture, flavour, taste, colour, and nutritional value of the product [19]. Lipid oxidation and lipid hydrolysis cause the generation of off flavours and loss of nutrients. Lipid oxidation increases after heat treatment due to the dramatic loss of antioxidant activity, as food antioxidants become ineffective during prolonged heating at elevated temperatures. Flavour changes in thermally treated meat products are caused by Maillard reactions, thermal degradation of lipids, and Maillard–lipid interactions [20]. Therefore, a compromise between food safety and minimum sensorial and physical properties must be reached during a thermal process [13]. 

To our knowledge, no study has been available that deals with the effect of different heat treatments and simultaneously with different fat levels in PLP. Thus, the objective of the present work was to investigate the influence of two heat treatments (including pasteurisation and sterilisation regimes) on the oxidative stability, colour, texture, and viscoelastic properties, as well as the microbiological and sensory quality of PLP with two different fat contents (30 and 40% (*w*/*w*)).

## 2. Materials and Methods

### 2.1. Preparation of Pâtés

The following ingredients were used for the production of PLP: shoulder lean pork meat, pork backfat, pork liver (all obtained from a local producer), broth, nitrite salt (NaCl with 0.5% (*w*/*w*) sodium nitrite), spices (mixture for liver pâté), and pork globin (also known as decolorised blood). Two different formulations of PLP were considered, differentiated in terms of fat content (30% (*w*/*w*) of fat (**P30**) and 40% (*w*/*w*) of fat (**P40**)). These fat levels were selected based on the common fat content in PLP and information available in the literature. The composition of both samples is presented in Table 1.

The pork meat and pork backfat were boiled for 120 min in water until tender (this liquid was later used as a broth in the recipe). Boiled meat, backfat, and hot broth were processed in a cutter (K 20 AC-6, Maschinenfabrik Seydelmann KG, Stuttgard, Germany) at 55 °C and 6400 rpm. At 45 °C, raw, ground pork liver with nitrite salt (K + S Czech Republic a.s., Prague, Czech Republic) was added and cut. Finally, a mixture of spices (RAPS GmbH & Co. KG, Kulmbach, Germany) and globin (VEOS NV, Zwevezele, Belgium) was added to the mixture and the cutting was terminated at a temperature of approx. 40 °C. The homogenous mass was filled into laminated aluminium cans (conical shape; inner dimensions of 26.8 mm height, 81.1 mm diameter at the top, and 68.9 mm diameter at the bottom; the weight of the sample in one can was approx. 95 ± 2 g) and the cans were sealed (NovaSeal–Nirosta Ltd., Chlumec nad Cidlinou, Czech Republic). Subsequently, the samples were divided into two groups: samples that were pasteurised (**PP**) and samples that were sterilised (**SP**). 

Samples designated for pasteurisation were thermally treated in a universal combi-oven (SelfCookingCenter^®^, SCC WE 61, Rational, Prague, Czech Republic) operating at 99 °C and 90–100% relative humidity. Target temperature in the geometrical centre of the container was 70 °C; this temperature was held for 10 min (temperature was controlled by applying a thermometer probe directly into the sample). Next, pasteurised pâtés were cooled down and cold-stored at 6 ± 1 °C. Pâtés selected for sterilisation were placed into autoclave (Fedegari FVA2/A1; Fedegari Autoclavi SpA, Albuzzano, Italy); sterilisation parameters were as follows: heating to target temperature (122 °C)—20 min, holding the sterilisation temperature—10 min, cooling to 50 °C—50 min. The actual temperature in the container placed at the coldest point in the retort was recorded using the Ellab TrackSense Pro dataloggers (Ellab A/S, Hilleroed, Denmark) and evaluated by the ValSuite 6.2 software (Ellab A/S, Hilleroed, Denmark). Subsequently, the sterilised pâtés were cooled and stored at room temperature (22 ± 2 °C). Both pasteurisation and sterilisation were performed the same day that the pâtés were manufactured. All analyses were realised after 1 week of storage. Three batches of PP and SP were produced. Therefore, 12 lots in total were made (2 fat contents × 2 heat treatments × 3 repetitions).

### 2.2. Microbiological Analysis

The microbial quality of the pâtés was evaluated by assessing the total number of aerobic and/or facultative anaerobic mesophilic microorganisms [21], the number of aerobic and anaerobic spore-forming microorganisms [22], and the number of yeasts and/or moulds [23]. Furthermore, a thermostat test was performed for SP samples [22]. All microbiological analyses were performed at least in triplicate and 3 packages were sampled (3 batches × 3 packages × 3 repetitions; *n* = 27).

### 2.3. Chemical Analysis

The dry matter, crude protein, and total lipid content were evaluated according to ISO 1442:1997 [24], ISO 1871:2009 [25], and ISO 1443:1973 [26], respectively. The pH value was measured with a pH meter equipped with a penetration probe (Edge™, Hanna Instruments, Prague, Czech Republic). The ammonia content was determined by Conway’s microdiffusion analysis [27]. The oxidative stability of the lipids was evaluated using the 2-thiobarbituric acid method as “TBARS-value” (thiobarbituric acid reactive substances) as described by Kristensen and Skibsted [28]. The results were expressed as absorbance units at the wavelength used per milligram of sample (A_538_/mg). All chemical analyses were performed at least in triplicate and 3 packages were sampled (3 batches × 3 packages × 3 repetitions; *n* = 27).

### 2.4. Textural Analysis

The texture properties of the pâtés were monitored using a texture analyser TA.XTplus (Stable Micro Systems Ltd., Godalming, UK) equipped with a 20 mm diameter cylindrical aluminium probe. Analysis was carried out by double compression of the sample within the original container (strain 50%, trigger force 5 g, deformation rate 1 mm/s, 6 ± 1 °C). Data collection was accomplished using Exponent Lite software (version 4.0.13.0; Stable Micro Systems Ltd., Godalming, UK). The following parameters were obtained from the force/time curves: hardness, cohesiveness, gumminess, springiness, and adhesiveness as described in Jooyandeh [29]. The force versus time data were converted to a corrected stress, Hencky strain, elongational viscosity, and Hencky strain rate using the following Equations:(1)$$\sigma_{C}=\frac{F\left(t\right)H\left(t\right)}{A_{0}H_{0}}$$
(2)$$\varepsilon_{H}=\ln\left(\frac{H_{0}}{H_{\left(t\right)}}\right)$$
(3)$$\eta_{E}=\frac{2F\left(t\right)H\left(t\right)}{\pi r^{2}v}$$
(4)$$\dot{\varepsilon_{H}}=\frac{v}{2H_{\left(t\right)}}$$
where σC is the corrected momentary stress (Pa), εH the dimensionless momentary Hencky strain, *η_E_* elongational viscosity (Pa·s), $\dot{\varepsilon_{H}}$ Hencky strain rate (biaxial extensional strain rate; s^–1^), *F*(*t*) the momentary force at time *t* (s), *H*_0_ the initial cylindrical sample height (m), *H*(*t*) the height (m) of the deformed sample at time *t* (s), *A*_0_ the cross-sectional area of the original sample (m^2^), *v* the velocity (deformation rate; m·s^–1^), and *r* is the radius of the sample (m) [30]. Textural analysis was performed in at least triplicate and 3 packages were sampled (3 batches × 3 packages × 3 repetitions; *n* = 27).

### 2.5. Rheological Analysis

A dynamic oscillatory shear rheometer (Rheostress 1, Haake, Bremen, Germany) was used to observe the viscoelastic properties of pâtés. Rheological analysis was performed within the linear viscoelastic region using a parallel plate–plate geometry (35 mm diameter, 1 mm gap) at 20.0 ± 0.1 °C. During analysis, the exposed edge of the geometry was covered with a thin layer of silicone oil to prevent the samples from dehydrating. The elastic (G′; Pa) and viscous (G″; Pa) moduli and the complex viscosity (η*; Pa·s) were determined at a frequency range of 0.01–10.00 Hz (amplitude of shear stress 20 Pa). Subsequently, the complex modulus (G*; Pa) and the loss tangent (tan δ) were calculated. G*, tan δ, and η* were presented at the reference frequency of 1 Hz. The rheological analysis was performed at least in triplicate and 3 packages were sampled (3 batches × 3 packages × 3 repetitions; *n* = 27).

### 2.6. Instrumental Colour Analysis

Colour measurements were performed using the HunterLab UltraScan^®^ Pro Color Measurement Spectrophotometer (Hunter Associates Laboratory, Inc., Reston, VA, USA). The CIE Lab colour scale with the illuminant D65 and 10° viewing angle was used. Parameters L* (luminosity/lightness; black to white), a* (chromaticity on a green to red axis/redness), and b* (chromaticity on a blue to yellow axis/yellowness) were determined according to the International Commission on Illumination. The calibration of the spectrophotometer was performed in reflectance mode, with specular reflection excluded, and using white (C6299) and grey (C6299G) reference plates. Chroma (C*), hue angle (H), whiteness index (WI), yellowness index (YI), and browning index (BI) were calculated according to Al-Hilphy et al. [31]. Colour analysis was carried out in at least triplicate and 3 packages were sampled (3 batches × 3 packages × 3 repetitions; *n* = 27).

### 2.7. Sensory Analysis

Twelve selected assessors and experts (who regularly consume pâtés) trained according to ISO 8586:2012 [32] were employed in the sensory analysis of pâtés. Appearance, consistency, and flavour were evaluated using a seven-point product quality scale (1—excellent, 4—good, 7—unacceptable; each point on the scale was objectively defined with quality parameters), whereas hardness and off flavour were assessed by means of a seven-point intensity scale (1—negligible, 4—medium, 7—excessive). The standard sensory laboratory equipped with sensory booths and fulfilling the requirements of ISO 8589:2007 [33] was used in the sensory assessment (normal light conditions, temperature 20 ± 2 °C).

### 2.8. Statistical Analysis

The results were evaluated using Kruskal–Wallis and Wilcoxon tests at a significance level of 0.05. The influence of (i) heat treatment and (ii) fat content was evaluated separately. Unistat^®^ 6.5 software (Unistat, London, UK) was used for statistical analysis.

## 3. Results and Discussion

### 3.1. Microbiological Analysis

The total number of aerobic and/or facultative anaerobic mesophilic microorganisms (e.g., *Microbacterium* spp., *Enterococcus* spp., *Staphylococcus* spp., *Micrococcus* spp., and *Rhodococcus* spp.) and the number of yeasts and/or moulds in PP were in the interval 1.70–2.40 and 1.65–2.18 log colony-forming units per gram of the tested sample (CFU/g), respectively (*p* < 0.05). P30 samples showed higher microbial counts than P40 samples (*p* < 0.05). These results agreed with those reported by Lorenzo et al. [1]. No aerobic or anaerobic spore-forming microorganisms were detected in the PP. Subsequently, none of the methods used, including the thermostat test, revealed the presence of any monitored microorganisms in the SP, regardless of the fat content of the sample. Hence, the applied sterilising mode (122 °C, 10 min) was sufficient for the inactivation of the microflora detectable by the methods used. From a commercial point of view, a food is commercially sterile if it is free of *Geobacillus stearothermophilus* or *Clostridium perfringens*. If a food is stored at higher temperatures (above 25 °C), sporulating thermophiles, such as *Clostridium botulinum* and *Clostridium sporogenes*, must also be eliminated, as they can pose a risk in countries with subtropical and tropical climate [13,14].

### 3.2. Chemical Analysis

The results of the proximate analysis (dry matter, crude protein, total lipid content, and pH values), ammonia content, and TBARS values are shown in Table 2. It can be seen that heat treatment did not alter either the dry matter, crude protein, and total lipid contents or the pH values (*p* ≥ 0.05). On the other hand, the different fat content resulted in significant changes in the proximate compositions of PLP. The P30 samples showed lower dry matter and total lipid content (*p* < 0.05) compared to the P40 samples. In contrast, P30 pâtés had a higher crude protein content (*p* < 0.05) in comparison with P40 pâtés. Reduction in dry matter and lipid content and simultaneously increase in protein content with decreasing fat content was also observed in the works of Lorenzo et al. [1], Lorenzo and Pateiro [9], Estévez et al. [34], and Delgado-Pando et al. [35]. These changes resulted from the composition of the raw material composition. The pH value is a fundamental factor during the emulsification process and affects both the physicochemical and functional properties of the emulsion. The pH value away from the isoelectric point can increase the shipment of proteins resulting in the enhanced effect of proteins during the emulsification process [6]. The pH values were not affected by the fat content (*p* ≥ 0.05), which is consistent with the work of Lorenzo et al. [1], Lorenzo and Pateiro [9], and Estévez et al. [34], who found no significant differences in pH in foal pâtés with various fat contents. 

Ammonia is considered an intermediate or final product of the reactions of nitrogenous compounds released during the course of the Maillard reaction complex and Strecker degradation of amino acids [17,36]. Consistent with this statement, an increase in ammonia amount of 22 and 32% was observed as a result of the sterilisation process in P30 and P40, respectively (*p* < 0.05; Table 2). A similar trend as a consequence of sterilisation treatment was described by Buňka et al. [27]. PLP with a lower fat content showed a higher ammonia concentration (*p* < 0.05), which is in agreement with the crude protein content observed, as more prominent deamination processes would be expected with a higher protein content.

Meat products are more susceptible to oxidation compared to fresh meat due to mincing and cooking processes that facilitate interactions between oxygen and free fatty acids, with heat and metalloproteins acting like a catalyst [11]. Malondialdehyde is claimed to be the most important biological breakdown product expected from lipid oxidation. The most commonly used method used in malondialdehyde quantification is the TBARS assay [37]. Sterilisation treatment resulted in a two-fold TBARS value (compared to pasteurised pâtés) regardless of fat content (*p* < 0.05; Table 2). Thermal treatment was previously described to significantly enhance the degree of lipid oxidation and, thus, leads to higher TBARS values [19,20]. Similarly, a higher fat content promoted lipid oxidation in PLP. The TBARS value in P40 was 41 and 35% higher compared to P30 after pasteurisation and sterilisation, respectively (*p* < 0.05). These results are expected since TBARS are derived from lipid oxidation and, thus, higher fat pâtés should imply higher amounts of oxidation products. Significant correlations between lipid oxidation and fat content were monitored also by Lorenzo et al. [1], Lorenzo and Pateiro [9], and Estévez et al. [34].

### 3.3. Textural Analysis

The texture of a food product is a series of physical characteristics resulting from their molecular, microscopic, and macroscopic structure, which can be classified as mechanical and rheological properties. Textural parameters describe changes in both elastic and surface properties and the internal structure of foods under analysis, as they reflect various physical phenomena (in the area of large deformations) occurring during the measurement [8]. The results of the textural properties (hardness, cohesiveness, gumminess, springiness, and adhesiveness) of pasteurised and sterilised PLP are presented in Table 3. As expected, different heat treatments had a significant effect on textural properties of liver pâtés. The sterilisation regime resulted in increased hardness, cohesiveness, gumminess and springiness, and decreased adhesiveness (*p* < 0.05). The greatest differences in textural properties of pasteurised and sterilised samples were detected in the case of hardness and gumminess (almost doubled values due to sterilisation), followed by cohesiveness (approx. 60% increase) and springiness (approx. 30% increase). Adhesiveness was lowered 3–6 times as a consequence of sterilisation. Raising hardness due to retorting of sausages was observed by Duranton et al. [38]. On the other hand, Rezler et al. [8] observed reduced hardness and adhesion in sterilised pork liver pâtés compared to pasteurised ones.

Advanced evaluation of results of the texture analysis (Figure 1) led to the conclusion that sterilisation caused significantly higher values (*p* < 0.05) of corrected stress (*σ_C_*; Pa) in most of the monitored range of Hencky strain (*ε_H_*; dimensionless) compared to the pasteurised samples. In Figure 2, a sharp increase in elongational viscosity (*η_E_*; Pa∙s) in the first part of the curve is clearly visible, corresponding to the transient flow regimes. This is followed by a practically linear part of the curve corresponding to the squeezing flow regime, which is independent of the increasing values of the Hencky strain rate ($\dot{\varepsilon_{H}}$; s^−1^). The linear part of the curve depicted in Figure 2 clearly showed that the elongational viscosity of PP was significantly higher (*p* < 0.05) compared to SP. For P30 pâtés, the latter-mentioned conclusion was valid also for the part of the curve that represents the squeezing flow regime.

The production of liver pâtés with diverse fat contents also resulted in different textural properties of the manufactured samples. Rising fat levels caused an increase in hardness, cohesiveness, gumminess, and springiness and a reduction in adhesiveness (*p* < 0.05), similarly to the sterilisation effect. Increased hardness may be conditioned by the density of segments of the spatial protein matrix and can be the consequence of changes in the internal structure of PLP occurring during the manufacture process [8]. Furthermore, adipocyte structure of backfat tissues can remain intact after the manufacture, and the fat which remains inside the adipocytes can contribute to the higher value of hardness [4,5]. Consequently, the values of corrected stress and elongational viscosity (Figure 1 and Figure 2) increased with increasing fat content (*p* < 0.05) across most Hencky strain values and Hencky strain rate values, respectively. Comparable findings, i.e., the decrease in hardness due to the decrease in fat content was observed by Martins et al. [4], Domínguez et al. [5], and Rezler et al. [8]. 

In contrast, Lorenzo and Pateiro [9] and Estévez et al. [34] stated that the presence of higher amounts of fat resulted in softer pâtés. Furthermore, Lorenzo et al. [1] and Lorenzo and Pateiro [9] described increased hardness, chewiness, gumminess, and cohesiveness as a consequence of fat reduction, which is not in agreement with our findings. These discrepancies can be explained by different compositions of the samples and different temperatures used during the textural analysis. Advanced evaluation of textural curves underlined and exactly supported the conclusions found during statistical analysis of common texture parameters (such as hardness, cohesiveness, gumminess, springiness, and adhesiveness). The application of the advanced textural data assessment approach to assess the effect of the heat treatment type and the fat content on the PLP in the large deformation area was not found in the available literature.

### 3.4. Rheological Analysis

Rheological measurements (in the area of the small deformations) were performed in the linear viscoelastic region of the PLP samples. The frequency sweep of the samples showed differences in their viscoelastic properties. The results of dynamic oscillatory rheometry for PLP are shown in Figure 3 and Table 3. It can be seen from the course of the G′ and G″ moduli in a frequency range of 0.1–10.0 Hz that a gradual increase in both recorded moduli was observed with increasing frequency (Figure 3). Both G′ and G″ moduli increased with increasing fat content and, also, as a result of different heat treatments (*p* < 0.05). P30 samples showed greater differences between pasteurised and sterilised pâtés compared to P40 samples. The decline in G′ due to the lower fat content was also observed by Rezler et al. [8]. However, the latter authors described reduction in G′ as a result of sterilisation, which is not consistent with our outcomes.

The G′ and G″ moduli can serve as valuable tools to express the intensity of elastic and viscous behaviour of viscoelastic materials [8,35]. The elastic modulus showed values higher than those of the viscous modulus in all samples (G′ > G″), thus implying that the pâtés possessed more elastic behaviour and acted like a gel with solid-type structures present. All of the samples showed typical viscoelastic behaviour with typical weak gel properties. This type of behaviour is usually shown by a 3D cross-linked gel network. The prevailing elasticity over viscosity in pork liver pâtés, liver paste, and meat emulsion was also described by Delgado-Pando et al. [35], Steen et al. [39], and Kumar et al. [40], respectively. Moreover, G′ and G″ are related to the strength of intermolecular interactions of the pâté emulsions and protein gel matrix, whereas the rate of change of these parameters is related to the stability of the matrix. The difference between G′ and G″ corresponds to the strength of the matrix [40]. In our study, this difference was greater in P40 samples (compared to pâtés with lower fat content), being the largest in SP40. Thus, this pâté showed stronger intermolecular interactions and a more stable matrix.

The values of G′ and G″ were related to the observed loss tangent values (tan δ; the ratio equal to G″/G′), which were less than 1 (Table 3), thus confirming the gel-like acting. Values of tan δ were not influenced either by fat content or by heat treatment (*p* ≥ 0.05). Furthermore, G* and η* were increased as a consequence of both sterilisation and increased fat level (*p* < 0.05). Raising values of G* and η* resulted in elevated firmness of the gel and corresponded with the hardness results obtained during the textural analysis (Table 3). Excellent correlations between the texture determinants and rheological parameters were also found in the study of Rezler et al. [8].

### 3.5. Instrumental Colour Analysis

Table 4 shows the results of the colour parameters (lightness, chromaticity on green to red and blue to yellow axes, chroma, hue, and colour indexes) of PLP. Colour analysis revealed that heat treatment significantly altered the colour parameters of all samples regardless of fat content. While the lightness of the pâtés was reduced as a result of the sterilisation process, both chromaticities increased (*p* < 0.05). Furthermore, sterilised pâtés had a more intense colour (higher C* values; *p* < 0.05), higher yellowness and browning indexes, and lower whiteness index (*p* < 0.05), which corresponds to the observed L*, a*, and b* values. Overall, sterilisation resulted in darker, yellower, and redder PLP as compared with pasteurisation. These results correspond to the findings described by Polak et al. [12] who studied sterilised and pasteurised chicken liver pâtés. Darkening of pâtés during sterilisation can be explained by the course of the Maillard reaction complex, since brown pigments, melanoidins, are formed [36]. Increasing browning intensity as a consequence of sterilisation was also observed by Xie et al. [20].

As the colour of pâtés is closely related to the colour characteristics of the raw material used for the manufacture [34], it is supposed that changes in the proportions of the ingredients would lead to different colour characteristics. This premise was confirmed, since a higher fat content (and thus a lower meat content) increased the lightness and reduced both chromaticities. The P30 samples were significantly darker, redder, and yellower (*p* < 0.05). The chroma and both the yellowness and browning indexes decreased as a result of higher levels of fat in PLP (*p* < 0.05), whereas whiteness index remained constant (*p* ≥ 0.05). Last but not least, the hue angle was not influenced by sterilisation or by different fat concentrations (*p* ≥ 0.05). Similar colour changes resulting from different fat levels in pâtés were also concluded by Lorenzo et al. [1], Lorenzo and Pateiro [9], Estévez et al. [34], and Delgado-Pando et al. [35].

### 3.6. Sensory Analysis

The effects of different heat treatments and fat contents on the sensory quality of pork liver pâtés are presented in Table 5. It is evident that heat treatment affected all sensorial properties evaluated regardless of fat content (*p* < 0.05). The appearance and consistency of SP deteriorated to very good or good compared to PP, which was evaluated as excellent or great. Flavour worsened even more considerably, from excellent or great to less good or poor (*p* < 0.05). The worsening of consistency and flavour as a consequence of sterilisation treatment can be linked to the observed results of hardness and off-flavour evaluation, since these parameters were more intense in sterilised samples (*p* < 0.05). These results also correspond to advanced levels of ammonia and TBARS that can be responsible for flavour deterioration [19,20]. The growing hardness due to sterilisation observed in sensory analysis was correlated with the results of the textural analysis. Furthermore, the worse appearance can be attributed to darkening of the pâtés during sterilization, which corresponds with the results obtained from the instrumental colour analysis. Observed changes in sensorial properties as a result of sterilization can be explained by the course of the Maillard reaction complex [20,41]. In accordance with our results, Rezler et al. [8] determined a more intensive foreign smell in liver pâtés due to sterilisation.

The different fat content also influenced the sensory quality of PLP, with the exception of appearance, which remained unchanged (*p* ≥ 0.05). P40 samples had better consistency and flavour and were harder (*p* < 0.05). Furthermore, SP40 pâtés were evaluated with less intense off flavour (*p* < 0.05). This was not observed in PP, since pasteurisation did not induce any off flavours (*p* ≥ 0.05). Better flavour and higher acceptance and palatability of pâtés with higher fat content were previously described [1,8].

## 4. Conclusions

In conclusion, both different heat treatments and fat content significantly affected various quality parameters of PLP. Sterilised samples and samples with higher fat level were more prone to suffer lipid oxidation, as measured by TBARS. Moreover, these pâtés showed an increase in hardness, cohesiveness, gumminess, and springiness, as well as in all determined moduli (G′, G″, and G*) and complex viscosity. Additionally, both samples that underwent sterilisation heating and included a lower fat content were darker, yellower, and redder, with more intensive colour, elevated yellowness and browning indexes, and decreased whiteness index. Furthermore, sterilisation caused worsening in appearance, consistency, and flavour, together with advanced off flavour perceived in these samples. The decrease in fat level resulted in a deterioration in consistency and flavour.

## Figures and Tables

**Figure 1 foods-12-02423-f001:**
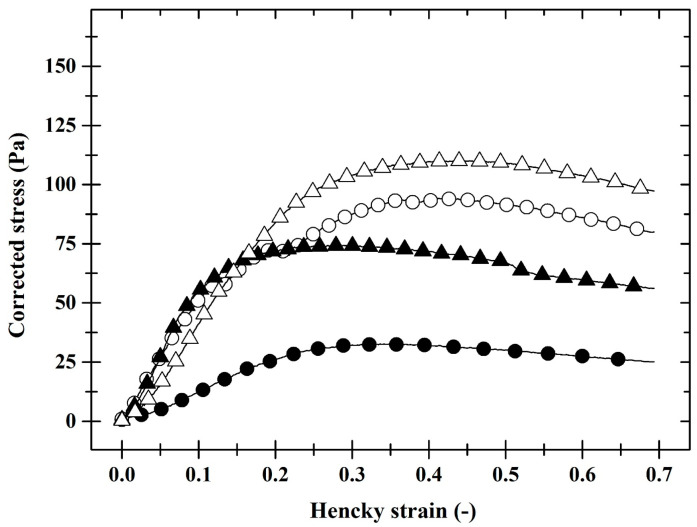
The dependence of corrected stress (Pa) on Hencky strain (dimensionless) of pasteurised (full symbols) and sterilised (open symbols) pork liver pâtés with fat content of 30% *w*/*w* (circle) and 40% *w*/*w* (triangle).

**Figure 2 foods-12-02423-f002:**
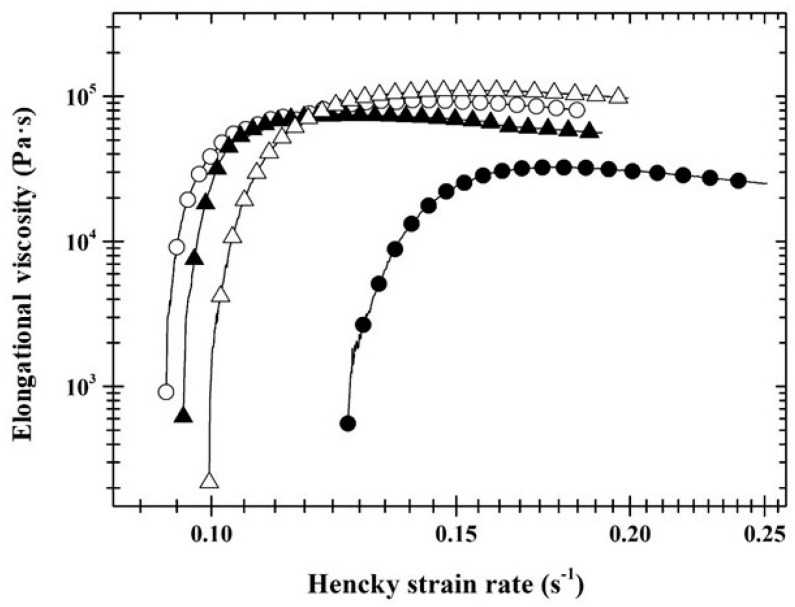
The dependence of elongational viscosity (Pa·s) on Hencky strain rate (s^−1^) of pasteurised (full symbols) and sterilised (open symbols) pork liver pâtés with fat content of 30% *w*/*w* (circle) and 40% *w*/*w* (triangle).

**Figure 3 foods-12-02423-f003:**
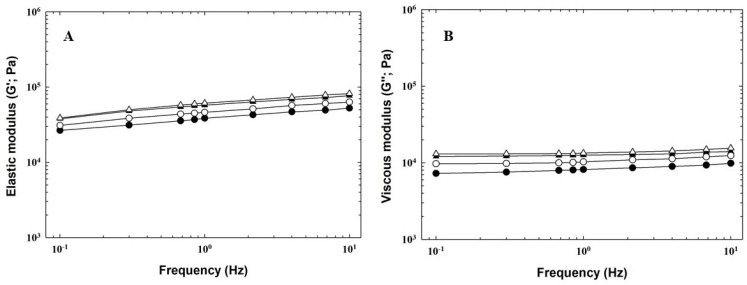
Dependence of the elastic modulus (G′; Part (**A**)) and the viscous modulus (G″; Part (**B**)) on the frequency (0.1–10 Hz) of pasteurised (full symbols) and sterilised (open symbols) pork liver pâtés with fat content of 30% *w*/*w* (circle) and 40% *w*/*w* (triangle).

**Table 1 foods-12-02423-t001:** Composition of raw materials (% *w*/*w*) used for manufacture of pork liver pâtés with content of fat 30% *w*/*w* (P30) and 40% *w*/*w* (P40).

Ingredients (%)	P30	P40
Pork meat	25	25
Pork backfat	30	40
Pork liver	22	22
Broth	20	10
Nitrite salt	1.5	1.5
Spices	1	1
Globin	0.5	0.5

**Table 2 foods-12-02423-t002:** Chemical composition of pasteurised and sterilised pork liver pâtés with content of fat 30% *w*/*w* (P30) and 40% *w*/*w* (P40). Results are presented as means ± SD (*n* = 27) *.

	Pasteurised Pâtés		Sterilised Pâtés	
	P30	P40	P30	P40
Dry matter (% *w*/*w*)	45.35 ± 1.06 ^a^ A	48.46 ± 1.46 ^b^ A	46.22 ± 1.27 ^a^ A	48.03 ± 1.16 ^b^ A
Crude protein (% *w*/*w*)	16.27 ± 0.32 ^a^ A	14.81 ± 0.24 ^b^ A	16.52 ± 0.28 ^a^ A	15.06 ± 0.29 ^b^ A
Total lipids (% *w*/*w*)	31.34 ± 0.70 ^a^ A	42.36 ± 0.91 ^b^ A	30.57 ± 0.65 ^a^ A	42.68 ± 0.92 ^b^ A
pH value	6.57 ± 0.05 ^a^ A	6.46 ± 0.04 ^a^ A	6.50 ± 0.08 ^a^ A	6.31 ± 0.07 ^a^ A
Ammonia (mg/kg)	52.13 ± 1.36 ^a^ A	43.07 ± 1.02 ^b^ A	63.47 ± 0.98 ^a^ B	56.67 ± 1.13 ^b^ B
TBARS (A_538_/mg)	39.73 ± 0.84 ^a^ A	56.34 ± 1.13 ^b^ A	82.62 ± 1.94 ^a^ B	111.86 ± 2.50 ^b^ B

* The means within a line (the effect of fat content) followed by different superscript letters (^a^ and ^b^) differ (*p* < 0.05). Pasteurised and sterilised samples were evaluated separately. The means within a line (the effect of heat treatment) followed by different capital letters (A and B) differ (*p* < 0.05). Samples with different fat content were evaluated separately.

**Table 3 foods-12-02423-t003:** Textural and viscoelastic properties of pasteurised and sterilised pork liver pâtés with content of fat 30% *w*/*w* (P30) and 40% *w*/*w* (P40). Results are presented as means ± SD (*n* = 27) *.

	Pasteurised Pâtés		Sterilised Pâtés	
	P30	P40	P30	P40
*Textural properties*				
Hardness (N)	2.07 ± 0.08 ^a^ A	3.29 ± 0.11 ^b^ A	4.49 ± 0.15 ^a^ B	5.64 ± 0.16 ^b^ B
Cohesiveness (-)	0.36 ± 0.01 ^a^ A	0.55 ± 0.01 ^b^ A	0.60 ± 0.02 ^a^ B	0.87 ± 0.02 ^b^ B
Gumminess (N)	1.14 ± 0.03 ^a^ A	1.73 ± 0.04 ^b^ A	2.26 ± 0.10 ^a^ B	3.32 ± 0.12 ^b^ B
Springiness (s)	5.03 ± 0.11 ^a^ A	6.11 ± 0.18 ^b^ A	6.63 ± 0.22 ^a^ B	7.72 ± 0.33 ^b^ B
Adhesiveness (-)	0.32 ± 0.01 ^a^ A	0.12 ± 0.00 ^b^ A	0.11 ± 0.00 ^a^ B	0.02 ± 0.00 ^b^ B
*Viscoelastic properties* **				
G* (Pa)	39,374 ± 1256 ^a^ A	58,748 ± 2067 ^b^ A	47,367 ± 1425 ^a^ B	62,664 ± 2179 ^b^ B
tan δ (-)	0.21 ± 0.01 ^a^ A	0.21 ± 0.01 ^a^ A	0.22 ± 0.01 ^a^ A	0.22 ± 0.01 ^a^ A
η* (Pa·s)	6266 ± 198 ^a^ A	8351 ± 246 ^b^ A	7539 ± 302 ^a^ B	9972 ± 326 ^b^ B

* The means within a line (the effect of fat content) followed by different superscript letters (^a^ and ^b^) differ (*p* < 0.05). Pasteurised and sterilised samples were evaluated separately. The means within a line (the effect of heat treatment) followed by different capital letters (A and B) differ (*p* < 0.05). Samples with different fat content were evaluated separately. ** Results of complex modulus, loss tangent, and complex viscosity are presented at the reference frequency of 1 Hz.

**Table 4 foods-12-02423-t004:** Colour parameters of pasteurised and sterilised pork liver pâtés with content of fat 30% *w*/*w* (P30) and 40% *w*/*w*(P40). Results are presented as means ± SD (*n* = 27) *.

	Pasteurised Pâtés		Sterilised Pâtés	
	P30	P40	P30	P40
Lightness (*L**)	66.67 ± 2.11 ^a^ A	61.76 ± 1.25 ^b^ A	60.46 ± 1.87 ^a^ B	54.73 ± 1.62 ^b^ B
Chromaticity (*a**)	11.75 ± 0.28 ^a^ A	10.21 ± 0.47 ^b^ A	12.83 ± 0.36 ^a^ B	11.35 ± 0.29 ^b^ B
Chromaticity (*b**)	16.25 ± 0.61 ^a^ A	14.44 ± 0.48 ^b^ A	17.31 ± 0.52 ^a^ B	15.71 ± 0.36 ^b^ B
Chroma (*C**)	20.06 ± 0.60 ^a^ A	17.69 ± 0.52 ^b^ A	21.54 ± 0.47 ^a^ B	19.38 ± 0.26 ^b^ B
Hue (*H*)	54.11 ± 1.57 ^a^ A	54.70 ± 1.83 ^a^ A	53.85 ± 1.61 ^a^ A	54.15 ± 1.42 ^a^ A
Whiteness index	57.62 ± 2.02 ^a^ A	57.86 ± 1.39 ^a^ A	52.97 ± 1.20 ^a^ B	51.40 ± 1.46 ^a^ B
Yellowness index	37.04 ± 1.44 ^a^ A	33.44 ± 1.03 ^b^ A	40.89 ± 1.57 ^a^ B	37.21 ± 1.62 ^b^ B
Browning index	42.77 ± 1.51 ^a^ A	37.90 ± 1.12 ^b^ A	48.16 ± 1.63 ^a^ B	44.27 ± 1.49 ^b^ B

* The means within a line (the effect of fat content) followed by different superscript letters (^a^ and ^b^) differ (*p* < 0.05). Pasteurised and sterilised samples were evaluated separately. The means within a line (the effect of heat treatment) followed by different capital letters (A and B) differ (*p* < 0.05). Samples with different fat content were evaluated separately.

**Table 5 foods-12-02423-t005:** Sensory evaluation of pasteurised and sterilised pork liver pâtés with content of fat 30% *w*/*w* (P30) and 40% *w*/*w* (P40). Results are presented as medians (*n* = 12) *.

	Pasteurised Pâtés		Sterilised Pâtés	
	P30	P40	P30	P40
Appearance	1 ^a^ A	1 ^a^ A	4 ^a^ B	4 ^a^ B
Consistency	2 ^a^ A	1 ^b^ A	4 ^a^ B	3 ^b^ B
Hardness	3 ^a^ A	5 ^b^ A	5 ^a^ B	6 ^b^ B
Flavour	2 ^a^ A	1 ^b^ A	6 ^a^ B	5 ^b^ B
Off-flavour	1 ^a^ A	1 ^a^ A	6 ^a^ B	4 ^b^ B

* The medians within a line (the effect of fat content) followed by different superscript letters differ (^a^ and ^b^) (*p* < 0.05). Pasteurised and sterilised samples were evaluated separately. The medians within a line (the effect of heat treatment) followed by different capital letters (A and B) differ (*p* < 0.05). Samples with different fat content were evaluated separately.

## Data Availability

The data used to support the findings of this study can be made available by the corresponding author upon request.

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
