# Peer review of "Influence of Heat Treatment on the Chemical, Physical, Microbiological and Sensorial Properties of Pork Liver Pâté as Affected by Fat Content"

_foods, 2023, doi:10.3390/foods12122423_

Round 1
Reviewer 1 Report
I am very grateful you for the invitation to review manuscript foods-2419845 by Lazárková and coauthors "Influence of heat treatment on the chemical, physical, microbiological, and organoleptic properties of pork liver pâté as affected by fat content”. This study aimed to evaluate the effect of heat treatment and fat content on selected properties of pork liver pâté. The work is interesting but needs adjustments to increase the quality of the material.
Comments:
- Abstract: Include a brief sentence about the importance of the study (problem to be solved).
- Line 18: How did that affect it? Please detail more descriptively.
- Lines 19-21: The sentence is generic. Include in detail the changes caused by heat treatments.
- Lines 22-22: What is the change caused by fat? This “basic” information should be better presented.
- Line 2; 23; 30; 76 and throughout the text: Change “organoleptical” to “sensorial”.
- Lines 25-26: Change the repeated keywords by different words from the title.
- Line 31: specify the “various types of meat”.
- Lines 34-37: Better specify the technological part associated with the preparation of the pâté.
- Introduction: What is the production and market for this type of product? These issues should be better presented.
- Lines 38-39: Please include in more detail the changes caused by the addition of fat.
- Lines 50-51: Indicate the effects for the sentence “are harmful to human health”.
- Lines 53-61: Further specify the process conditions generally applied to this type of products.
- Lines 109-113: Standardize the units used throughout the text (min; minutes).
- Line 199: Indicate more precisely the most abundant and problematic microorganisms for this type of product.
- Line 225: Please discuss in detail the role of pH in the preparation of emulsions (biochemically and technologically).
- Lines 268-270: Describe more precisely how increasing temperature affects the indicated parameters.
- Lines 294-296: How does the increase in fat concentration contribute to the increase in hardness, for example? (contrary to what was observed by the featured authors).
- 3.5. Instrumental colour analysis: It is not clear how the heat treatment changes the color. Describe in more detail.
Reviewer 2 Report
The manuscript is well written and presents some interesting findings oon fat and temp combination in pork liver pate. The hypothesis is well explianed and sound. The language is clear and easy to understand.
I have following observations
i. Please mention the level of significance in abstract in L 20-23; plz brifly mention whether these changes were desirable or not?
ii. L45-46: plz rewrite
iii. L58-59: plz add reference for the claim
iv. L74: loss of antioxidant capacity- please mention the cause
v. L88: plz mention source of pork globin
vi. L89-90: Please mentio the reason for the selection of fat levels? Based on reliminary trials or available literatures
vii. Fat has been added by replacing broth or pork; as the formulation in both should follow the same trend; I prefer the broth (which seems by the Table 1), so meat remain the same for better comparision of the results
viii. Table 1: please check the fat level in P40; as I think typographically it is wrongly written as 50 in place of 40
ix. L112-116: packaging for sterilization? Can or pouch and specify them
x. The sample size is sufficient to replicate the study
xi. L119-120: analysis id done after 1 week storage?
xii. Table 2: plz mention clearly in footnote that small capital letter indicates---and capital alphabet---
xiii. L256: Plz mention whether the TBARS value is within the prescribed limits
xiv. Table 3: increase in fat level increases the hardness? Please add suitable reasoning for the same. May be better emulsion formation or better gel strength etc; plz add after L 318-19
xv. Table 5: No SD or SE with mean
